# RadPhysBio: A Radiobiological Database for the Prediction of Cell Survival upon Exposure to Ionizing Radiation

**DOI:** 10.3390/ijms25094729

**Published:** 2024-04-26

**Authors:** Vassiliki Zanni, Dimitris Papakonstantinou, Spyridon A. Kalospyros, Dimitris Karaoulanis, Gökay Mehmet Biz, Lorenzo Manti, Adam Adamopoulos, Athanasia Pavlopoulou, Alexandros G. Georgakilas

**Affiliations:** 1DNA Damage Laboratory, Physics Department, School of Applied Mathematical and Physical Sciences, National Technical University of Athens (NTUA), Zografou Campous, 15780 Athens, Greece; vaso.zn@gmail.com (V.Z.); spkals@central.ntua.gr (S.A.K.); gkymhmt@gmail.com (G.M.B.); 2Department of Life Sciences, University Paris-Saclay, Saint-Aubin, 91190 Paris, France; dimitrispapak@gmail.com; 3School of Electrical and Computer Engineering, National Technical University of Athens, 15780 Athens, Greece; dkaraoul@gmail.com; 4Naples Italy and Radiation Biophysics Laboratory, National Institute of Nuclear Physics (INFN), Section of Naples, Department of Physics “E. Pancini”, University of Naples Federico II, 80138 Naples, Italy; manti@na.infn.it; 5Department of Medicine, Medical Physics Laboratory, Democritus University of Thrace, 68100 Alexandroupolis, Greece; adam@med.duth.gr; 6Izmir Biomedicine and Genome Center (IBG), 35340 Balcova, Izmir, Turkey; athanasia.pavlopoulou@deu.edu.tr; 7Izmir International Biomedicine and Genome Institute, Dokuz Eylül University, 35340 Balcova, Izmir, Turkey

**Keywords:** database, ionizing radiations, radiobiology, biophysical model, machine learning

## Abstract

Based on the need for radiobiological databases, in this work, we mined experimental ionizing radiation data of human cells treated with X-rays, γ-rays, carbon ions, protons and α-particles, by manually searching the relevant literature in PubMed from 1980 until 2024. In order to calculate normal and tumor cell survival *α* and *β* coefficients of the linear quadratic (LQ) established model, as well as the initial values of the double-strand breaks (DSBs) in DNA, we used WebPlotDigitizer and Python programming language. We also produced complex DNA damage results through the fast Monte Carlo code MCDS in order to complete any missing data. The calculated *α*/*β* values are in good agreement with those valued reported in the literature, where *α* shows a relatively good association with linear energy transfer (LET), but not *β*. In general, a positive correlation between DSBs and LET was observed as far as the experimental values are concerned. Furthermore, we developed a biophysical prediction model by using machine learning, which showed a good performance for *α*, while it underscored LET as the most important feature for its prediction. In this study, we designed and developed the novel radiobiological ‘RadPhysBio’ database for the prediction of irradiated cell survival (*α* and *β* coefficients of the LQ model). The incorporation of machine learning and repair models increases the applicability of our results and the spectrum of potential users.

## 1. Introduction

It is well documented that ionizing radiation (IR), in the form of electromagnetic waves or particles, is capable of causing a wide variety of DNA damage that is spatiotemporally correlated to irradiated cells, ranging from single-strand breaks (SSBs) and base damage to double-strand breaks (DSBs) and DNA cross-links. In the case that the burden of damage overwhelms the repair capacity of the cells, IR, similarly to other potentially carcinogenic agents such as non-ionizing radiation, may eventually lead to genetic instability, excessive cell death or cancer-promoting processes [1]. More specifically, the induced clustered DNA damage is considered to have a high probability of initiating carcinogenesis [2]. Therefore, modern anti-cancer radiation therapy (RT) faces the challenge of exerting or potentiating its deleterious effects on cancer cells, for example, by increasing the complexity of DNA damage, while decreasing the probability of sub-lethally damaging the surrounding healthy cells and tissues [3]. Hadrontherapy, with protons or ^12^C ions, enables a more targeted administration of IR and the selective destruction of cancer cells.

In recent decades, radiobiological models, together with Monte Carlo (MC) algorithms, such as PARTRAC [4], Geant4-DNA [5,6] and KURBUC [7], represent the main research approaches to estimating IR-induced damage at the molecular level with the closest possible approximation. Many radiobiological models developed to assess cell response to IR, such as the microdosimetric kinetic model (MKM) [8] and the local effect model (LEM) [9], have been used thus far to estimate cell survival upon exposure to IR. Of those models, the most widely applied is the linear-quadratic (LQ) model [10]. Recently, efforts have been made to integrate machine learning (ML) techniques in order to improve models’ adequacy and efficacy in terms of predicting dose distribution or forecasting treatment response [11].

In the field of radiobiology, there are a few databases and resources available that provide valuable information on various aspects of the biology of the irradiated cells. These sources contain data on radiation-induced biological effects, radiation dosimetry, DNA damage and repair kinetics, treatment plans and related areas. One notable radiobiology database is PIDE [12]. Such databases can serve as valuable resources for researchers, radiobiologists, clinicians and radiotherapists, since they provide access to a large volume of data that contribute to our understanding of radiation’s biological effects and their applications in cancer treatment. However, they usually consider only a single type of radiation (e.g., either particle or electromagnetic radiation), and they contain only two parameters regarding the type of biological response, i.e., the two LQ model coefficients (*α* and *β*).

Herein, we developed a computational biophysical model, which is able to accurately predict the response of human cells (i.e., conduct an assessment of complex DNA lesion and cell survival) after exposure to different types of IR at various dose levels. We were mainly motivated by the current lack, in the existing literature, of a ‘complete’ radiobiological model for the prediction of these critical types of biological response. To this end, we applied a fast Monte Carlo code (Monte Carlo damage simulation (MCDS)) [13] for predicting the average number of complex DNA lesions, such as DSBs, non-DSBs and SSBs, in the cells irradiated with γ-rays, protons, α-particles and carbon ions, by providing as input data the dose of each specific type of radiation. Our scope was the construction of a reliable radiobiological database which includes the literature-derived experimental data complemented by the MCDS simulation results for the induced lesions, as well as the development of an ML biophysical model/prediction tool. Our ultimate goal is to provide a user-friendly, publicly accessible resource for conducting research or facilitating clinical applications like radioprotection or radiotherapy.

## 2. Results

### 2.1. Experimental Data Calculations

As reported in the literature, early-responding tissues are known to have *α*/*β* ratios of around 7–10 Gy, while late-responding tissues have *α*/*β* ratios of 3–5 Gy (Table 1) [10,14]. Of note, the tabulated values are largely based on X-ray data; thus, a direct comparison against all radiation qualities (Figure 1) has some inherent limitations.

Obviously, these values are approximate and may vary depending on the specific study, patient population, and treatment techniques used. To be more precise, α and β are heavily scattered when comparing cells of the same tissue and even when analyzing the same cell line in different labs, growth conditions, etc. The *α*/*β* ratio is not a fixed property of a tissue, but rather represents an average value used for treatment planning.

The results for the dependence of *α*/*β* values from cancerous tissues and all types of radiation are shown in Figure 1a,b. The data have been ordered in two grouped box charts, where the majority of the values seemingly range from 4 to 6. The abbreviations ‘Panc’, ‘ConTis’, ‘HnN’, ‘NerTis’, ‘PleEff’ and ‘UmbCor’ refer to Pancreas, Connective Tissue, Head and Neck, Nerve Tissue, Pleural Effusion and Umbilical Cord, respectively. For this process, the negative *β* values were omitted, and only *α*/*β* values up to 20 were taken into account. More specifically, for the generation of the values, 8 datasets were used for the bladder, 4 datasets for the gastric, 15 datasets for the blood, 20 datasets for the bone, 295 datasets for the brain, 4 datasets for the kidney, 62 datasets for the breast, 40 datasets for the pancreas, 107 datasets for the colon, 4 datasets for the uterus, 8 datasets for the connective tissue, 5 datasets for the head and neck, 2 datasets for the muscles, 4 datasets for the nerve tissue, 27 datasets for the liver, 224 datasets for the lung, 62 datasets for the prostate, 6 datasets for the sacrum, 87 datasets for the skin, 10 datasets for the thyroid, 2 datasets for the eye and finally 1 dataset for the pleural effusion, the foreskin and the umbilical cord.

The correlation between the linear coefficient *α* of the LQ model for all types of radiation and LET is shown in Figure 2. Each point represents the average *α* value at the specific LET. We focused on the LET values ranging up to 150 keV/μm, where the majority of *α*-values fluctuated until 3.50–4.0 Gy^−1^. As is largely confirmed by the literature, *α* appears to have a nearly linear increase with LET, up to 100–150 keV/μm [15,16]. In this study, we fitted the data in the Origin 2018-64bit software through a linear equation, as shown in Figure 2; the exported R-Square (R^2^) values are shown in Figure 2d.

Due to the way survival studies are designed, and the nature of LQ, errors are not uniformly distributed as a function of survival, with typically smaller (absolute) uncertainties at the lower survival levels.

The dependence of DSBs on LET, which is depicted in Figure 3, is also of great radiobiological importance, since the abovementioned complexity increases with the density of ionization events. There is substantial evidence supporting that by increasing LET, the yields of DSBs increase up to an LET value of 150–200 keV/μm [4]. In this work, we focused on those LET value ranges of up to 150 keV/μm, at which the majority of the DSB values fluctuated. In Figure 3, the orange squares represent the experimental data collected from the relevant literature, while the blue diamonds indicate the output data from the MCDS simulation. Each data point corresponds to the average value of DSBs per Gray per Gbp for the five aforementioned radiation types for each LET value.

Based on the mathematical model for DNA repair kinetics, our results indicate good fitting between the experimental and predicted values (Appendix A). Moreover, by comparing our results to these of [17], we may observe an agreement among the repair percentages. As seen in Figure 8 of the mentioned article, for 100 keV electrons, this means that for the majority of our collected data, their repair percentage at 2 h, for example, is around 40%, while ours varies at around 50%. All our analytical kinetics *k_i_* parameters are included in Appendix A. This allows users to apply this mathematical model according to the initial DNA damage values and make predictions for a type of cell line (fibroblasts, epithelial and lymphocytes) regarding the expected remaining DSB value, for example, at a time point of 24 h post-irradiation. 

### 2.2. Model Predictions

The actual values of *α* and *β* are plotted on the *y*-axis, while the ML model predictions are plotted on the *x*-axis in Figure 4.

The permutation variable importance [18,19], a model-agnostic method, provided a better understanding of the impact that different variables have on the coefficients α (Figure 5a) and β (Figure 5b). More specifically, these diagrams depict which features are more critical for the prediction of each coefficient.

## 3. Discussion

The LQ model, since its formulation about fifty years ago, has been the dominant radiobiological model, as it provides an accurate and easily implemented description for the vast majority of the available, experimentally obtained, cell survival dose–response relationships. It makes use of the exponential relation (Equation (1)) to predict cell survival, where the two coefficients *α* and *β* are often referred to as “radiosensitivity parameters”, since they exhibit a strong dependence on the irradiated tissue. It is assumed that the *α* term represents the direct cell killing (‘single hit’), that is, lethal damage caused by a single incident particle, while the *β* term represents the impact of cell killing from ‘multiple hits’ [20]. By understanding the relationship between radiation dose and cell survival, the LQ model enables us to determine the most effective strategy for radiation treatment so as to maximize tumor control while minimizing normal tissue complications.

If the plot of cell survival versus dose is depicted on a log scale, a quadratic response curve is created. In the first part of the curve at low doses, the linear *α* term prevails, while the curvature increases as the quadratic *β* term becomes more significant. It is noteworthy that the *α*/*β* ratio essentially defines the degree of curvature, as well as the type of tissue which is being irradiated, serving as a measure of sensitivity to fraction size. It is fairly well-established that organs which contain rapidly proliferating cells are less sensitive to the fraction size of a dose. These early-responding tissues are known to have high *α*/*β* ratios, around 7–10 Gy, suggesting that higher doses delivered in fewer fractions can be more effective in controlling such tumors. Conversely, organs with slower cellular turnover exhibit much greater sensitivity to the dose fraction size. Late-responding tissues have been shown to have *α*/*β* ratios of 3–5 Gy [10], indicating that they are more effectively treated with conventional fractionation, where lower doses are delivered into multiple fractions over several weeks. Determining the appropriate *α*/*β* ratio for a specific tissue or tumor type is a crucial step for treatment planning in radiation therapy in terms of optimizing the radiation dose and fractionation schedules to achieve maximum tumor control while minimizing side effects on the surrounding healthy tissues.

Intratumoral heterogeneity greatly affects tumor response to radiotherapy, since the tumor is composed of cells that are both sensitive and resistant to radiation. More specifically, the vast majority of our data are considered asynchronous, which means that the cells in the cell culture of the experiments are in a different phase of their cycle, and therefore show different degrees of radiosensitivity [21]. In this way, when irradiating a heterogeneous cell population, the sensitive cells are expected to die at lower doses than the more resistant cells; this process may cause an ‘upward-bending’ survival curve, corresponding to a negative *β* value in the LQ model, and a positive second derivative [22]. 

The LQ model is an empirical, biologically based model, which means that it is specifically designed to provide an in-depth description of the radiobiological effects of cell killing and sub-lethal repair. Moreover, it uses very few parameters, which allow the model to be relatively simple in use, and also its predictions about cell killing up to 18 Gy are in agreement with the majority of the radiobiological mechanistic models [23]. On the other hand, at doses below 1 Gy, cells die from excessive sensitivity to small doses of ionizing radiation but become more resistant to larger doses. As a result, the model underestimates, in this region, the biological effect of a given dose. In general, the LQ model’s simplified representation of a linear and a quadratic term may not accurately capture the complexity of biological responses to radiation, as cellular responses can involve non-linear mechanisms and complex interactions between different pathways.

Apart from the LQ model, various, similar, well-known radiobiological models are available. For example, the Repair–Misrepair (RMR) model [24] and the Lethal–Potentially Lethal (LPL) model [25] are used extensively in the published studies. Furthermore, in the case of estimating radiation quality, such as heavy ions with high RBE values (e.g., ^12^C), more specialized models are needed, such as the Local Effect Model (LEM) [9] and the Microdosimetric Kinetic Model (MKM) [8]. The latter approaches go one step further and delve into the nanoscale deposition of energy and its impact on radiation sensitivity [10]. However, apart from their differences in structure and mechanism of operation, it is noteworthy that the LQ model provides almost equal predictions for the basic quantities to the aforementioned ones [23], even though it may not include as many features. In this respect, the LQ model could be considered as practical and easy to use, since the other models have many more parameters, which makes their use more complicated.

Regarding the results for the *α*/*β* values, as depicted in Figure 1a,b, they are in very good agreement with the values reported in the literature, as shown in Table 1. Interestingly, the kidney and lung have values of around 5, while the bladder has values of around 2. Furthermore, the *α*/*β* values for the bones and the skin are around 7, whereas for the head, neck and colon, the values are around 6. There might be slight deviations, however, as the experimental values of *β* are often very small, dramatically affecting the corresponding ratio. The frequency of the tissue also plays an important role, since some tissues occur more frequently, resulting in more reliable data, while others appear less. It should be mentioned that the direct comparison between in vitro- and in vivo-derived LQ values is likely not directly meaningful. All our LQ values data are based on cellular data.

Regarding the dependence of the *α* term of the LQ model from LET, the data follow almost the same scattering pattern in every case (Figure 2), so there is not a great dependence of *α* on the cell type. As far as R^2^ is concerned, although the values from Figure 2d reveal a positive correlation, they are not particularly high, as the data represent the average *α* values of all types of radiation for each LET value. By taking into consideration the collected data, different radiation types of the same LET exert diverse biological effects, and therefore have different linear terms [26]. Although two types of radiation may have the same LET, their different characteristics and mechanisms of interaction with biological tissues may lead to varying biological effects. The data are also produced, using experimental measurements, from different laboratories with different equipment, a different dose range and perhaps under different oxygen and dimethyl sulphoxide (DMSO) conditions. However, due to the fact that R-square is a limited metric for very heterogeneous data, as it is sensitive to outliers, we also calculated the Spearman correlation, which is a metric that is less sensitive to outliers. According to Figure 2d, the value of the metric is 0.71 for normal cells, 0.75 for tumor cells and 0.74 for both normal and tumor cells. Given that the closer the Spearman correlation coefficient is to 1, the better the correlation between data, there is an apparently strong, positive linear correlation between LET and the *α* coefficient.

There is a general upward trend in the theoretical results, that is, the ones from the MCDS simulation (Figure 3). However, many values exist between 6 and 8; this is probably due to the fact that different radiation types of the same LET have different linear terms [26], as mentioned above. It has been demonstrated that the linear term is linked to lethal damage and subsequently to DSBs, which means that, for the same LET, different radiation types lead to different numbers of DSBs. This upward trend of values is anticipated, according to the relevant literature. As the LET increases, the number of DSBs induced per cell and the complexity of the breaks are expected to increase [27]. More specifically, high-LET radiation tends to produce more complex DNA damage, including a higher number of DSBs, as compared to low-LET radiation like γ-rays or X-rays. This happens because high-LET radiation deposits more energy per unit distance, leading to a greater likelihood of simultaneous DNA strand breaks. This fact has also been demonstrated in the literature through the use of other simulation tools like PARTRAC [28,29].

On the other hand, regarding the experimental DSB values, an increase occurs in relation to LET, followed by a corresponding decrease. As the data points represent the average yields of all radiation types, the pattern is not clear enough. Early experimental studies on DSB induction through radiation with different qualities have shown that DSBs tend to increase with increasing LET, while other, later, studies have demonstrated a reduction in DSBs with increasing LET, most probably due to experimental detection weaknesses [30]. In this case, it is presumed that the indirect strand breaks decrease as the LET value increases, while the directly induced ones remain constant and then also decrease at a very high LET. Regarding the decrease in the direct effects at a very high LET, it is considered that the extra energy is wasted at the same breakage that has already occurred [31]. Of particular note, the relationship between LET and DSBs is not linear and can be influenced by other factors; for instance, the repair mechanisms within cells can vary depending on the type of radiation and the cellular context.

Furthermore, the majority of experimental values are lower than the theoretical ones in all diagrams in Figure 3. This disparity was expected, as there is an inherent difficulty in the measurement of experimental damage. More specifically, the techniques followed for damage calculation include manual handling or automatic processes by evaluating the total γH2AX immunofluorescence intensity emission per cell, using high-throughput techniques such as flow cytometry. These methods are often not sensitive enough, which makes them not efficient for this purpose [32,33]. Thus, since foci detection techniques rely on microscopy, the quantification of experimental damage is associated with advances in fluorescent microscopy technologies [34]. This could also explain the abovementioned decrease in DSBs, which might be present, but would be difficult to measure with the current techniques. Of particular note, Monte Carlo simulations are mathematical models which can provide useful predictions, though they are not exact replicas of experimental measurements. Their accuracy in predicting DSBs depends on several factors, including the accuracy of the underlying physical models, the quality of the input parameters and the data, as well as the complexity of the system that is simulated.

In the plots of the actual versus the predicted values for *α* and *β* (Figure 4), the closer the dots that correspond to the pairs (actual; predicted) fall to the diagonal (black) line, the better the predictive power of the model. A linear least squares method was applied to model these points and then the red line was plotted over the same plots. The results show a good predictive performance for *α* coefficient (R^2^ = 0.63), whereas the prediction of *β* was not that satisfactory (R^2^ = 0.6), as we expected, for the reasons mentioned above. This can also be explained by the scale of *β* values and the experimental process that is used to extract *α* and *β* values. These values are calculated by fitting the LQ model on the survival curves for each model. This process itself introduces an error, which, in the case of *α,* is not significant in relation to its mean magnitude, but is significant in relation to *β*. More specifically, there is intrinsic noise, to the point where it is difficult to predict *β* in an adequate way. This conclusion can be also extracted from the performance of the model in predicting *α* and *β*, by using the root mean square error (RMSE) of actual versus predicted values from the test dataset. The metric RMSE, which is the square root of the average of squared errors, was used to measure the accuracy of the model, that is, the differences between the predicted and observed values. This is always non-negative, and the closer the RMSE values are to zero, the better the model’s prediction. The RMSE value and the Spearman correlation coefficient for *α* were calculated as 0.55 and 0.72, respectively, indicating a satisfactory prediction for *α* in combination with the value of R^2^. Concerning *β*, the RMSE value and the Spearman correlation coefficient were calculated as 0.24 and 0.4.

Finally, both the features ‘LET’ and ‘irradiation Conditions’ were found to be important for the prediction of *α* and *β* coefficients (Figure 5). The importance of the effect of LET was expected [11], as the *α* and *β* coefficients (indicative of the lethal and potential lethal damage) are drastically affected by the energy that is deposited per distance. As regards the irradiation conditions, that is, whether the radiation is mono-energetic or spread-out Bragg peak (SOBP), the difference in the energy distribution on the target cell appears to have a different effect on lesions, drastically affecting both *α* and *β*. The fact that cell line variance effect is present but low may reflect the existence of various contaminated cell lines by other cultures. For example, the HSG line is now known to actually be a HeLa derivative [35] (https://www.cellosaurus.org/CVCL_2517 accessed on 4 April 2024). Providing consensus cell line names in the database is not possible in all cases, since there is not a universally accepted name list. More specifically, some HeLa-contaminated lines are mentioned, such as HEp-2 and L132, and it is stated that ECV-304 is a derivative of T24 [36]. However, many cell line origins are still in question, requiring further investigation. The users can exploit the radiobiological data in RadPhysBio based on the original data and information provided and can draw their own conclusion.

In this work, we analyzed thousands of datasets of original experimental radiobiological data and calculated the predictive efficiency of *α* and *β* parameters for cell survival with the use of the widely accepted LQ model. Based on these data, we developed an original, publicly available database, RadPhysBio, which includes several physical and biological parameters. Our results show meaningful and valuable predictions for most of the data (*α*-values). Nevertheless, we acknowledge that this study has several limitations and uncertainties due to the wide variance in the original experimental data used and of the software selection. The difference in *α* between different cell lines is less than that of the most extreme LET conditions. This is attributed to the fact that there are quite significant differences amongst cell lines, particularly for low-LET X-rays, which should not be neglected in any predictive model.

## 4. Methods

### 4.1. Data Collection

Initially, we collected the experimental irradiation data of human cells for five different types (Linear Energy Transfer (LET)) of radiation: X-rays, γ-rays, carbon ions, protons and α-particles. We manually searched the bibliographic database PubMed [37] from 1980 until 2023 and we correspondingly created five different data files, one for each radiation type. Then, we stored these data in a database.

Specifically, we used the relevant keywords [“X-ray” OR “γ-ray” OR “carbon” OR “proton” OR “alpha particles”] AND “radiation” AND “human cells” to search the published studies for data relevant to the corresponding five types of radiation. After we inserted the keywords, we recorded only the publications which included information about the α,β coefficients or/and DNA damage. Regarding X-rays, we collected 522 publications out of 11,634 results, from which we recorded 1098 experiments. In the same way, for γ-rays, we collected 277 publications out of 6611 results, obtaining 506 experimental datasets (EDs). For carbon ions, we collected 276 publications out of 2394 results, from which we recorded 676 EDs. For protons, we gathered 163 publications out of 1339 results and obtained 393 EDs. Finally, for α-particles, we collected 53 publications out of 1018 results, from which we recorded 100 EDs. Therefore, in total, we collected 1291 publications out of 22,996 results, from which we recorded 2773 experimental datasets (EDs).

Overall, this database combines various physical and biophysical characteristics of radiation and enables estimation of the induced biological damage through coefficients used by the LQ model, as well as quantification of the DSB and non-DSB clusters.

A workflow of the process is illustrated in the Figure 6 below:

Regarding γ-rays, we assumed that the values for the relative biological effectiveness (RBE) and LET are both almost equal to 1. For X-rays, we also assumed that the RBE value is almost 1; the LET value, as reported in the literature, for an energy range around 250 kVp, is equal to 2 keV/μm, while for energy values between 6 and 15 MeV, it is equal to 0.3 keV/μm. For an energy range less than 100 kVp, we assumed that LET is approximately 2.5 keV/μm. For the other types of radiation (carbon ions, protons and α-particles), LET was calculated from well-known RBE/LET data [38] using the WebPlotDigitizer software (https://automeris.io/WebPlotDigitizer.html, accessed on 9 September 2023) with a maximum estimated uncertainty of 20%.

In brief, the LQ survival model provides a simple relationship between cell survival *S* and the delivered dose *D*. In this model, the fraction of the surviving cells is equal to the sum of a linear term *αD* and a quadratic term *βD*^2^:(1)$$S=e^{-\alpha D-\beta D^{2}}$$

In order to calculate the *α* and *β* coefficients of the model, in case the latter were not provided in the corresponding publication or an experimental Survival-Dose diagram was provided instead, we used WebPlotDigitizer software [39], which allowed us to extract numerical data from those plot images (for more details, refer to Appendix A). In particular, we analyzed the raw survival data from each diagram without considering extreme values, and then we fitted those pairs with the theoretical equation of the LQ model through the Python code [40] in the Spyder environment [41] of Anaconda3 [42], in order to obtain *α* and *β* coefficient values (refer to Appendix A for more details). More specifically, from each survival curve, we included at least three pairs in the fitting process. Subsequently, we included the obtained values of both coefficients from the publications, together with the fitted ones (wherever needed), in the new database.

For the calculation of the initial values of the induced DSB or non-DSB clusters (in case they were not provided in the publications), we again used the WebPlotDigitizer software. From the experimental Damage–Dose or Damage–Time plots, we extracted the initial number of DSBs or non-DSB clusters per Gray per Giga base pair (bp), within the time period of a quarter or half an hour after the irradiation. If the damage was calculated, for example, per cell per two Grays, we divided the given number by 6.4 (as we assumed that the size of DNA in a single cell is ~6.4 Gbp) and by 2. In the majority of the collected papers, the number of the induced γH2Ax foci was reported instead of the DSBs, and thus we assumed a one-to-one ratio between them and included them in the count. Therefore, in the present database we considered both damage values from the corresponding publications (337 experimental values) and the calculated ones, when necessary. We also recorded each initial damage in an Excel spreadsheet, where “p” indicates that the damage was calculated with the pulsed-field gel electrophoresis method [43] and “f” denotes damage calculation with the foci method [44].

Regarding the use of the Monte Carlo simulation tool, MCDS, we developed the RadPhysBio database in response to the worldwide demand for a comprehensive radiobiological database, which combines cell survival data with DNA damage data and radiation’s physical parameters (e.g., energy, LET and dose rate). The lack of specific DNA damage data in cell survival papers has raised the necessity of using a Monte Carlo simulation code. Because of this, we included such data, being aware of the limitations of MCDS as well as other Monte Carlo codes. The data from MCDS have proven to be—in this work and previous works—quite compatible with the experimental data [45,46] based on the selection of several parameters, like oxygen percentage, energy, scavenging capacity, etc. According to the authors’ experience in applying ML algorithms [11] and vast knowledge of experimental data [30], there is often a great variability in the DNA damage values, which can even vary by one to two orders of magnitude when obtained from different laboratories. Therefore, we deem that the MCDS data can be considered a good indicator for users of the level of damage expected in each case, and at the same time cannot be considered as a variable input forcing the model in a specific direction.

For protons, carbon ions and α-particles, we performed the induced DNA damage simulation using this code by inserting into the input file the specific energy of each experiment and the oxygen concentration (i.e., 20% for normoxic cell conditions and 1% for the hypoxic ones); for X-rays and γ-rays, we inserted in the input file two different energy values (1 keV and 10 keV) and the previously mentioned values for oxygen concentration. From the corresponding table provided in the output file of MCDS (refer to Appendix A for more details), the magnitudes which are important to report in our work are the average total number of DSBs and the average number of non-DSBs (‘Other’). The former is the number of the induced DSBs per Gray per Giga base pair (Gbp) for every type of radiation that we mentioned above, and the latter is approximately the number of non-DSB clusters per Gray per Gbp.

The features of the database are presented in detail in the following Table 2.

### 4.2. DNA Repair Fitting Model

As far as DNA repair is concerned, we used the NHEJ model with 9 variables and 10 unknown rate constants [47] and reproduced the process. We presented the change in dose equivalent Deq(Gy) with time (h) for lymphocytes, fibroblasts and epithelial cells (Appendix A) according to this model using MATLAB 2019b software [48]. For each type of cell line, a table of the experimental and theoretical data is also provided (Appendix A). In order to obtain the mean values of the k parameters which best fit to the data, we calculated the average values of DSBs for each value of k and then we numerically simulated them. The sum y_i_ (for i = 1 to 9) represents the number of the unrepaired cells. In our calculations, the dose rate was set to dD/dt = 80 Gy/h and the induction rate per unit dose constant was set as a = 0.2. We also present the experimental and calculated fitting repair data (Appendix A), which contain the number of initial DSBs per Gy per cell (we assumed that the size of DNA in a single cell is ~6.4 Gbp, even though DNA content differs from cell to cell, depending on the phase of the cell cycle) and the remaining DSBs% with post-irradiation time, based on the data in the original publications. In addition, all critical derived k parameters of the NHEJ model per tissue category were included. In addition to the existing initial repair data in the database, some extra repair data were added (for lymphocytes and epithelial cells), and these are symbolized with “+”, as they are not included in the database but only in Appendix A.

### 4.3. Database Development

For the creation of the RadPhysBio database, the following process was employed: 1. Data Preparation: Excel files containing diverse datasets were gathered and standardized. Node.js scripts were crafted to merge these datasets into a common JSON format using libraries like ‘xlsx’ and ‘fast-glob’. This step ensured the uniformity and accessibility of the data for the web application. 2. Web User Interface Development: The web user interface was developed using Nuxt.js, a powerful framework that offers server-side rendering for improved SEO and enhanced initial load times. Web Workers were harnessed to execute resource-intensive tasks in the background. This approach prevents UI blocking and ensures a smoother user experience. Dexie.js was utilized as a client-side database library and it allowed for the efficient storage and retrieval of data, reducing the need for frequent server requests and thereby enhancing overall performance. To expedite the UI development process, Tailwind CSS was adopted. This utility-first CSS framework provided a responsive and mobile-friendly design system, facilitating rapid prototyping and consistent styling. 3. Hosting and Deployment: Netlify (https://www.netlify.com/ accessed on 5 July 2023) was chosen as the hosting platform for the web application.

### 4.4. Machine Learning and Prediction Model

Regarding the ML algorithms, we gathered all radiation data in one Excel spreadsheet (containing 2774 rows) and we added an extra column “RadiationType” in order to differentiate the type of radiation. We filled the missing LET values based on (a) existing RBE–LET curves for a known RBE value [38,49], (b) the MCDS-calculated results in case the energy value was given, and (c) similar *α* values in case the RBE and energy values were not provided. As far as DNA damage is concerned, we filled in the columns with the experimental values, and when they were not recorded, we filled the blanks with the values from our simulations. In the case of X-rays and γ-rays, we chose the energy value to be 1 keV, which is considered an average value. It should be noted that we used the value “1” for DMSO concentration, which stands for 1 mol/dm^3^.

In this study, the problem we are dealing with is multivariate regression with categorical and continuous independent variables and a considerable amount of missing values. To address all these issues, like in our previous work [11], randomForestSRC 3.2.3 [50] software in the R programming language was used, which is designed for regression and classification purposes. The random forest algorithm [51] has also been adapted to handle survival data and multivariate regression tasks. The overall results indicate that the highest performance is achieved when *α* and *β* are predicted together, in contrast to other models applied to predict *α* and *β* separately, underlying the complex interaction between *α* and *β*.

The following conceptual pipeline was employed: (1) In an effort to increase the likelihood of achieving good prediction efficiency, the dataset consisting of 2773 samples was divided into training (90%; 2257 samples) and testing (10%; 251 samples) datasets. More specifically, from the 2773 data, we used the 2508 with a non-zero LET value. We also included results for 70–30% training–testing datasets (70%, 1755 samples, and 30%, 753 samples) in the Appendix A, although there was not a significant statistical difference between their performance. (2) We tuned the model so as to find the optimal hyperparameters:-mtry: Number of variables to possibly split at each node.-nodesize: Minimum size of the terminal node.

The tuning of the model was executed by repeatedly fitting the random forest model with multiple combinations of the mtry and nodesize values and calculating the Out-Of-Bag (OOB) error. We chose those values that corresponded to the minimum OOB error; in our case, mtry = 7 and nodesize = 1 (for 70–30% mtry = 9 and nodesize = 1).

(3) We trained the model with the optimal hyper-parameters (for more details, refer to Appendix A). (4) We used the optimal interpretations, as follows:-Performance (qq plots).-Variable importance [51].

## 5. Conclusions

In this work, we present RadPhysBio, an original radiobiological open-access database, combining thousands of datasets of cell survival along with DNA damage values and prediction, using MC code and ML. The future goal of this research is the enrichment of the database with additional experimental data, as well as with new parameters, such as cell death, in order to improve the biological modeling. Furthermore, the development and application of meta-heuristic optimization algorithms such as genetic algorithms is of great importance in order to improve the fine-tuning of model parameters. These algorithms have the ability to search and converge, even in huge search spaces, for optimal sets of fitting parameters [52].

## Figures and Tables

**Figure 1 ijms-25-04729-f001:**
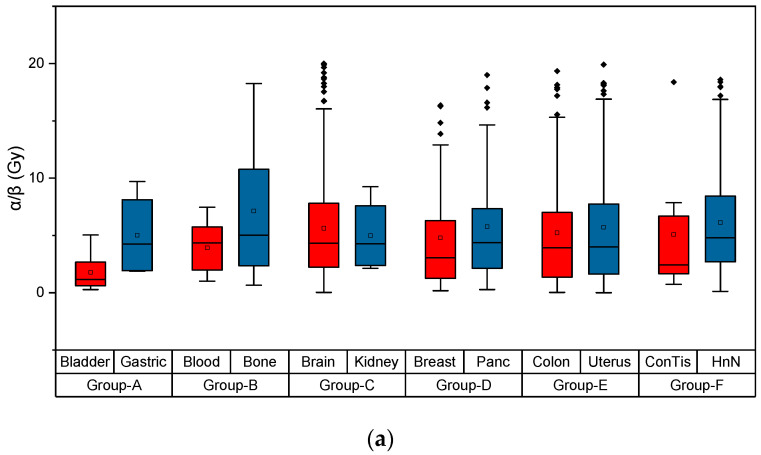
Grouped box charts for different sets of tissues, as explained above (**a**) bladder-gastric, blood-bone, brain-kidney, breast-pancreas, colon-uterus, conntissue-HnN and (**b**) muscles-nerve tis, liver-lung, prostate-sacrum, skin-thyroid, eye-pleEff, Foreskin-UmbCord, for the *α*/*β* ratio for the different types of cancerous tissue. The sample size included almost 996 datasets.

**Figure 2 ijms-25-04729-f002:**
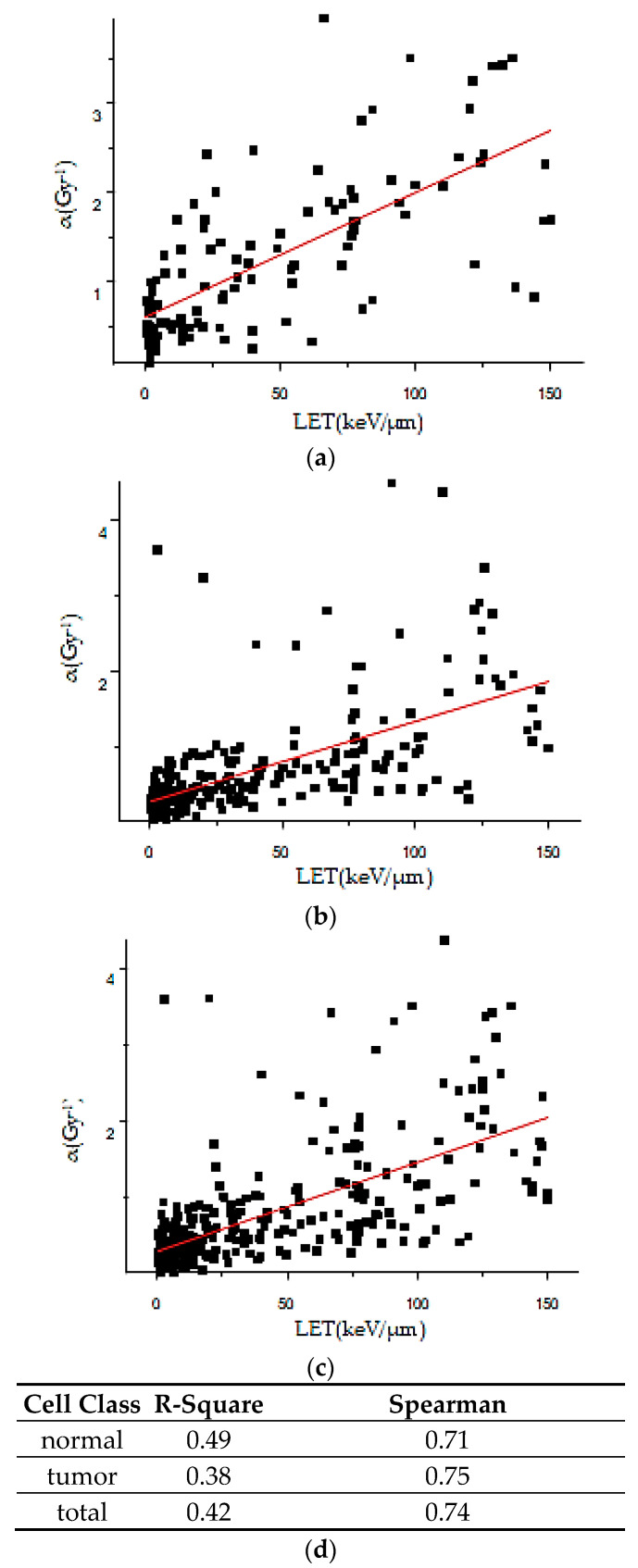
Plot of *α* versus LET for: (**a**) normal cells; (**b**) tumor cells; (**c**) both normal and tumor cells, where the red line is the optimal line passing through the pairs. R-square of the fitting process per cell class category is presented in (**d**). The sample size included 697 normal cell datasets and 1960 tumor cell datasets, while both normal and tumor cell datasets totaled 2657.

**Figure 3 ijms-25-04729-f003:**
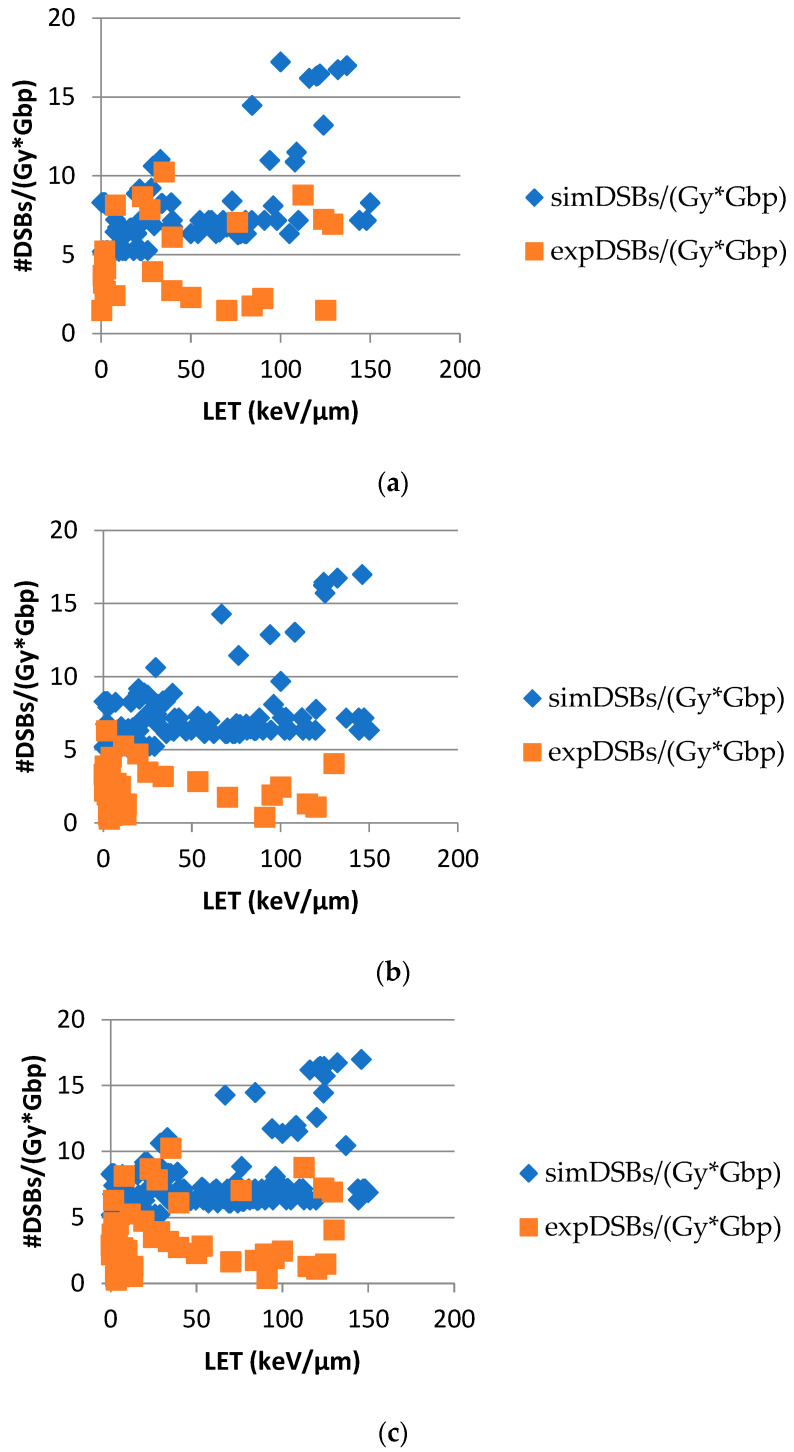
DSBs per Gy per Gbp with LET for (**a**) normal cells, (**b**) tumor cells and (**c**) both normal and tumor cells. The sample size included 683 normal cell datasets and 1881 tumor cell datasets, while both normal and tumor cell datasets totaled 2564.

**Figure 4 ijms-25-04729-f004:**
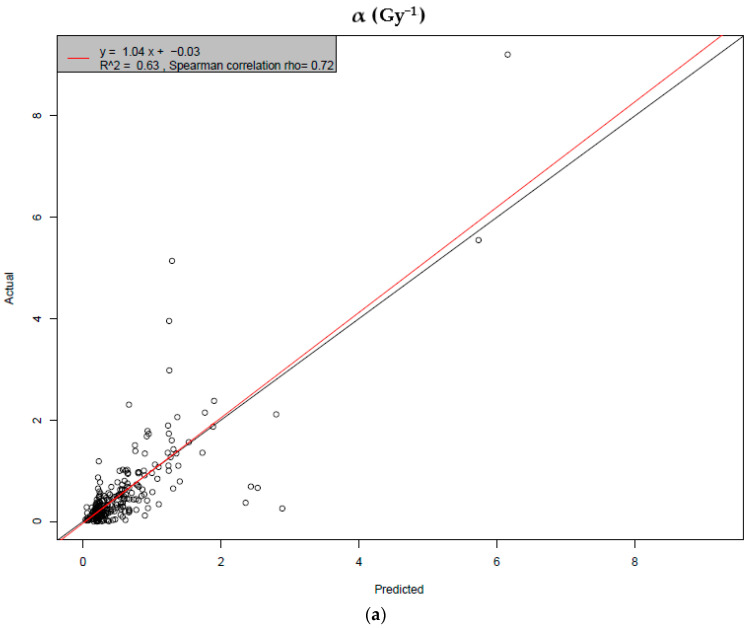
Plots of the distribution of *α* and *β* values against the distribution of their predictions: (**a**) *α* performance; (**b**) *β* performance. The red line represents the distribution function of the pairs (actual; predicted), while the black line represents the identity function ‘y = x’. The closer the points lie to the diagonal line, the better the model’s performance. All 2773 datasets were used for this procedure.

**Figure 5 ijms-25-04729-f005:**
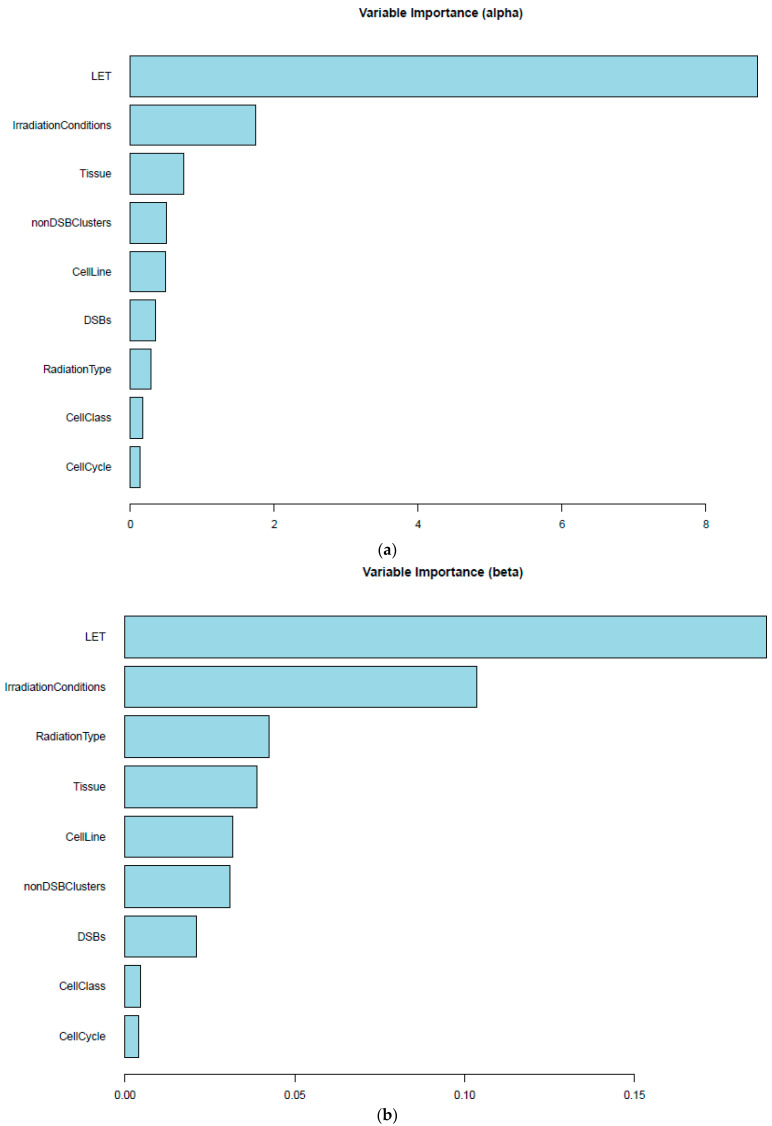
Permutation variable importance. The effect of each variable on the Out-Of-Bag (OOB) error during model training for (**a**) *α* and (**b**) *β*. The larger the change in the OOB error, the greater the importance of the given feature. All 2773 datasets were used for this procedure.

**Figure 6 ijms-25-04729-f006:**
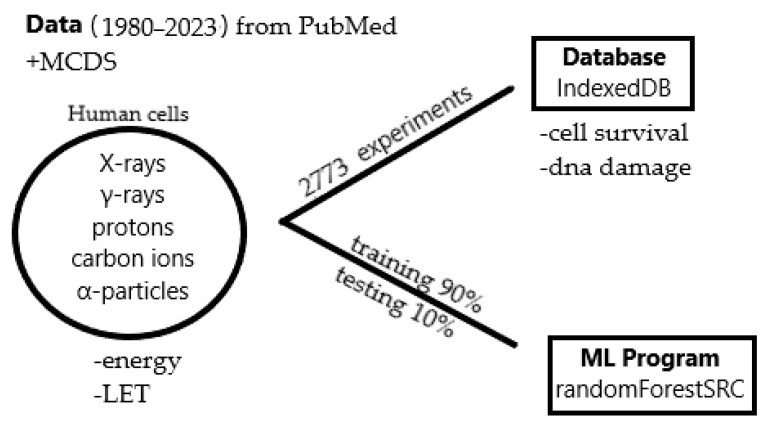
Presentation of the workflow diagram.

**Table 1 ijms-25-04729-t001:** Literature-derived suggested *α*/*β* value ranges per tissue.

Tissue	Range of *α*/*β*
kidney	Late-responding tissue 3–5
lung	
bladder	
bone	Early-responding tissue 7–10
head and neck	
colon	
skin	

**Table 2 ijms-25-04729-t002:** A detailed presentation of the features of the database.

Column	Content
#ExpID	Running number labelling the database entry
PMID	Running number labelling the publication
#Exp	Running number labelling the irradiation experiments within a publication
CellLine	Name of the irradiated cell line
Tissue	Name of cell tissue
CellClass	Tumor cells (t) or normal cells (n)
CellCycle	Cell cycle phase (phases are provided explicitly in each case, or noted by ‘a’ for ‘asynchronous’ cell lines)
Source	Type of radioactive source
Energy (MeV)	Specific radiation energy
RBE	Relative Biological Effectiveness
LET (keV/μm)	Linear Energy Transfer in water
IrradiationConditions	Mono-energetic radiation (‘m’), or spread-out Bragg peak (‘s’)
DoseRate (Gy/min)	Quantity of radiation delivered per minute of time
*α*	Linear coefficient of the LQ model (in Gy^−1^) for response to radiation, as given in the corresponding publication, or else from fitting to raw data
*β*	Quadratic coefficient of the LQ model (in Gy^−2^) for response to radiation, as given in the corresponding publication, or else from fitting to raw data
DSBs/(Gbp*Gy)	Number of initial DSBs per Gbp per Gy, as given in the corresponding publication
nonDSBClusters/(Gbp*Gy)	Number of initial non-DSB clusters per Gbp per Gy, as given in the corresponding publication
DSBs_1%O2	Number of initial DSBs per Gbp per Gy, calculated by the MCDS simulation code, for the specific energy of each experiment and 1% oxygen concentration in the cell
Other_1%O2	Number of initial non-DSB clusters per Gbp per Gy, calculated by the MCDS simulation code, for the specific energy of each experiment and 1% oxygen concentration in the cell
DSBs_20%O2	Number of initial DSBs per Gbp per Gy, calculated by the MCDS simulation code, for the specific energy of each experiment and 20% oxygen concentration in the cell
Other_20%O2	Number of initial non-DSB clusters per Gbp per Gy, calculated by the MCDS simulation code, for the specific energy of each experiment and 20% oxygen concentration in the cell
1keV_DSBs_1%O2	Number of initial DSBs per Gbp per Gy, calculated by the MCDS simulation code, for 1 keV energy and 1% oxygen concentration in the cell
1keV_Other_1%O2	Number of initial non-DSB clusters per Gbp per Gy, calculated by the MCDS simulation code, for 1 keV energy and 1% oxygen concentration in the cell
1keV_DSBs_20%O2	Number of initial DSBs per Gbp per Gy, calculated by the MCDS simulation code, for 1 keV energy and 20% oxygen concentration in the cell
1keV_Other_20%O2	Number of initial non-DSB clusters per Gbp per Gy, calculated by the MCDS simulation code, for 1 keV energy and 20% oxygen concentration in the cell
10keV_DSBs_1%O2	Number of DSBs per Gbp per Gy, calculated by the MCDS simulation code, for 10 keV energy and 1% oxygen concentration in the cell
10keV_Other_1%O2	Number of initial non-DSB clusters per Gbp per Gy, calculated by the MCDS simulation code, for 10 keV energy and 1% oxygen concentration in the cell
10keV_DSBs_20%O2	Number of initial DSBs per Gbp per Gy, calculated by the MCDS simulation code, for 10 keV energy and 20% oxygen concentration in the cell
10keV_Other_20%O2	Number of initial non-DSB clusters per Gbp per Gy, calculated by the MCDS simulation code, for 10 keV energy and 20% oxygen concentration in the cell

## Data Availability

Publicly available datasets were analyzed in this study. These data can be found here: http://radbiodb.physics.ntua.gr/radphysbio/ accessed on 7 April 2024.

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
