# Peer review of "RadPhysBio: A Radiobiological Database for the Prediction of Cell Survival upon Exposure to Ionizing Radiation"

_ijms, 2024, doi:10.3390/ijms25094729_

Round 1

Reviewer 1 Report (Previous Reviewer 1)

Comments and Suggestions for Authors

As the authors can probably surmise, I have recently reviewed this manuscript for another journal. While I was glad to see they had addressed some of my concerns, this prompted me to review the database in more detail which highlighted what appears to me to be serious issues in the data quality in almost every field. While the manuscript appears to still be under review in that journal, I assume this is an issue with their editorial system, and the authors are resubmitting here post rejection or withdrawal.

However, the extensive comments I provided to that version of the manuscript do not seem to have been considered at all, so I will simply reproduce most of them below for completeness, with a few additions based on some of the authors' comments in the response letter which I had not previously seen.

While I acknowledged the amount of work the authors have put in in all my reviews of this manuscript, I must again underscore that just because something was time-consuming is not enough to merit publication on its own. And given the numerous issues with the applicability of this data, I could not recommend this database in its current state.

As noted above, the following are adapted from my review of this manuscript elsewhere, with some light editing to reflect comments in response to review.

------

With regards to concerns about specific fields:

- Cell Line/Tissue:

As emphasised in previous comments, the authors have used the cell line name as reported (including typos and in some cases experimental numbering, as in Furusawa et al PMID 11025645). There are two issues with this - firstly, it makes it more difficult to identify what studies used the same cell lines, as e.g. PC3 and PC-3 cells would be treated as different cells in an analysis which did not resolve this naming discrepancy, as the authors' appears to. This would hugely reduce the power of the database to detect cell-line specific trends. A number of samples also have a cell line which is only described in generic terms, e.g. 'fibroblasts', despite likely being derived from different sources.

In addition, even amongst the same cell line there are differences in the tissue of origin. Some of these are synonyms which may be relatively benign, only affecting the power but not significantly impacting conclusions (e.g. fibroblasts vs skin for skin fibroblasts), others are fundamentally different (e.g. Colo679 is identified as both colon and skin cancer), which will affect any analysis, and U937 is noted as both blood and pleural effusion, and as normal tissue and cancer in two different entries. Similarly, the only examples of foreskin tumour cells are mislabelled AGO1522 cells, and from a skim over the database there are many similar issues - simple counts in excel suggests ~2 dozen cell lines with identical names marked as tumour and normal tissue, and ~65 cell lines reported as being from different tissues. This is likely an under-estimate, as it doesn't consider differences where there's also typographical differences in cell line name (e.g. MOLT4 vs MOLT-4).

And I remain convinced that it is very significant that a large number of cell lines are now well-known to be contaminated by other cell lines, often to the point of never having existed. Most notably for high-LET studies, the widely used HSG line (138 samples in the database) is a HeLa derivative. This must be indicated, as it fundamentally alters the interpretation of much of this data (being in a very different site, and a different biology to that proposed for HSG). If the authors do not have a relevant resource, I would recommend Cellosaurus (https://www.cellosaurus.org/index.html) which is a widely respected database of cell line characteristics, and includes extensive annotation of both recommended naming conventions, and identification of contaminated cell lines.

Consistency of data is essential for large scale analyses, and all of these issues would fatally undermine any attempt to perform cell- or tissue-specific analyses in these data. The authors have previously raised concerns about noting different lines to the groups which carried out experiments, but this can be resolved by splitting this into 'reported' and 'consensus' names and tissue of origin columns, together with a thorough review of the data for consistency.

- Photon Radiation (MeV), Energy (MeV) and LET (keV/μm):

From a brief review of these fields, there again seems to be poor consistency in reporting, to the point where I am unsure what each value is meant to be.

For X-ray sources, "Photon Radiation (MeV)" appears to variously either be the maximum energy of the photon field used, or in some cases simply the maximum energy of the machine, regardless of the energy which is used in the experiment. The latter is obviously not so informative for an experimental condition, as machines can easily be run lower than this. However, for about half of photon points, this is simply not reported, and "Energy (MeV)" is reported instead (and approximately 10% of data reports both).

For the energy field, there is again ambiguity - although it is described as "Specific radiation energy, evaluated at the target", in many cases this simply appears to be the peak energy of the source again as a single value (although this is not always the case, without clarity about why). However, this is not a good approximation of the true energy of the radiation seen by the cells, both due to the radiation characteristics of the source and target geometry. So it's unclear how this corresponds to the field name, both in terms of 'specific energy' and 'at the target' (the latter of which I would usually interpret to mean the cell target, which is the only other usage of 'target' within this manuscript). Clarity on the actual nature of both of these fields, and a more accurate quantification of target energy, would be invaluable to evaluate any photon energy dependence.

This effect is even more critical for heavy particles. As noted above, at first reading it seems that the Energy value represents the energy when the beam reaches the cell target. However, it appears that in almost no case is this the value used, and instead this is most commonly the primary energy of the beam, typically at the exit of the source. Radiobiologically, this energy is not particularly interesting, as beams are degraded as they go through targets and phantoms, and the variation of this energy is very significant radiobiologically. Comparing the initial peak energy to biological effects at a fraction of that energy, without a link to target depth and geometry is not a particularly meaningful choice, and the database does not provide information to resolve this.

However, even allowing for this being the peak energy, its recording is still not consistent - for some datasets (e.g. PMID 17007551), simply the mean initial beam energy is reported. However for some cases with SOBP only the highest energy component of the spread-out beam is quoted (e.g. 31194786), while in others the midpoint of the SOBP energy range is quoted (PMID: 32347564). The combination of somewhat biologically irrelevant energy combined with inconsistent reporting also makes it unlikely these data could be usefully used.

This could be resolved somewhat by using the LET instead, but this is also not without problems. Firstly, it is highly unclear how LET is obtained - the authors mention one approach in the methods, but three in the results. Notably, two of these (Jones RBE model fitting, and simply matching a nearby alpha value) introduce circular dependencies into the analysis. If the authors assume a particular LET-RBE relationship exists in the data, then it is very unsurprising that they obtain a strong LET-RBE relationship in their subsequent analysis. The use of the Jones model is further undermined by the fact that the quantitative value of the LET-RBE relationship is far from agreed! To be useful for predictive modelling, LET values should only be obtained from calculations or measurements independent of the biological endpoints (i.e. Monte Carlo or similar dose deposition simulations) otherwise they are no longer independent variables and cannot meaningfully be used for prediction.

Furthermore, many of the LET values seem to suffer from similar issues to the Energy field, such as for PMID 17007551 and 27380803, where the authors report the same energy and LET for a range of target depths, apparently based on the initial energy. In reality, across these depths the particle energy is decreasing and LET increasing, as can be seen in the escalating biological effect. By not accurately incorporating this variation in energy and LET, further essential data is lost.

- RBE:

The authors extensively discuss RBE, but at no point do they clearly define what definition of RBE they are using (i.e. comparison at a fixed dose, fixed survival, something else?). This is important as RBE is known to be both endpoint (see below on alpha and beta) and survival level dependent. Similarly, some RBE values seem to be taken directly from papers, while others (e.g. 31194786) seem to be recalculated, but it's unclear if this is to a different dose level, or because the authors have recalculated some other parameter. More clarity is needed here.

- Alpha and beta:

I am glad that the authors were able to resolve the issues with previous alpha and beta parameters. However, on re-inspection of these data, I noticed a further issue with the types of data being collected. Specifically, looking at the very lowest sensitivities, I notice the authors have used several datasets (e.g. 11153144, 24411610, 23868054) which did not use clonogenic assays, but rather metabolic (MTT) or dye exclusion assays. It is well-known that these assays report much higher viability than the clonogenic assay (e.g. as reported in 11153144, SF2 for MTT was ~0.97 compared to 0.68 for clonogenic assays). While both types of assay are informative, this huge difference in dose range and sensitivity means they cannot be meaningfully combined in a single analysis (which similarly affects RBE predictions in these lines also). Due to other issues with their interpretation I'd typically recommend removing all of the non-clonogenic assays from the database entirely, or at a very minimum including a column noting the assay type for survival studies so they can be analysed independently.

- MCDS data:

It remains fundamentally unclear to me the purpose of the MCDS modelling in this dataset.

For all of the 1keV_ and 10keV_ data, an identical value is presented for every photon exposure, and NA for every non-photon exposure. This provides zero additional physical information that's not provided by the radiation type field. The DSBs_* column likewise is purely a function of the beam energy and particle type, which again provides no additional information to a fit beyond those other columns. Moreover, because as noted above the beam energy in the database does not reflect that seen by target cells, in many cases this data predicts identical damage yields for cells seeing very different radiation types, potentially further misleading analyses if included.

Moreover, even if these data were robustly calculated, their value as a predictive tool is unclear, as it is focusing heavily on one model's predictions, which have previously been fit in other datasets. Thus it's unclear they can be used for any type of discovery analysis, and will bias the analysis to favour trends originally incorporated in the MCDS.

In their response and most recent revisions, the authors actually seem to be suggesting that the MCDS values would actually be preferred over experimental data, which seems to raise the question about why they're doing any of this database generation at all, since if they have full confidence in MCDS it can also be used to generate RBE predictions?

I would recommend deleting all of these data, to clarify the distinction between experimental and simulated values in this database, or at the least more clearly distinguishing between these points (e.g. clearly marking them as SIM or similar so an inattentive user doesn't think they're real data).

- General:

A major challenge with interpreting all of the data in this database is that it is a mix of values predicted by the original experimental paper authors and values re-fit during the generation of the database, through various methods. I would strongly recommend consistent reporting approaches, i.e. having columns which only contain as-reported values from the authors, supplemented by other columns with database-generated parameters; or systematically refitting all survival parameters and recalculating LETs etc. in a single consistent framework.

Failing either of those, clearly denoting which values are original and which are refit would significantly improve the database.

I fully appreciate, as the authors have raised at many points, that many of these steps are a lot of work. (I am also somewhat concerned that the raw data has been lost, which presents significant challenges around traceability and updatability!) But many of the values presented in this database at present are not representative of the underlying experimental conditions at all, and simply leaving fields empty would be a more appropriate solution than including somewhat incorrect values, or those imputed from other sources with limited tracability.

As a final note, some text in the database appears to be in Greek (particularly some cell line names starting with tau, which is indistinguishable visually from a Latin capital T), which would further complicate programmatic analysis of cell line names and tissues of origin.

I also have some concerns about the manuscript text, but it's hard to know how much of that stems from the above issues, so I won't go into those at this stage.

Comments on the Quality of English Language

-

Author Response

Reviewer 2 Report (New Reviewer)

Comments and Suggestions for Authors

This is an interesting study and the data is a welcome addition to the existing data, which are scattered among many publications and are not easy to access.

Major comments

1.    It is not easy to understand why the machine learning tool is needed for this study. It appears the authors used this technique to estimate the alpha and beta value from published experimental data. The data must be survival data. How was the data used with the random forest model? The authors could use a data-fitting or regression algorithm. It appears this is based on the authors’ work published in Ref.[7]. If so, it should be cited in Section 4.4.

Minor comments:

[line 53] The meaning of “precise administration of IR” is unclear. In other words, the precise administration is not limited to protons and heavy ions.

[Figure 4] What are red and black lines? You explain it in lines 321-324 but add the description in the figure caption.

[line 232] The LQ model does not overestimate but underestimates cell survival. Ref.[16] does not mention high dose cases. You meant Ref.[19], which concludes that the LQ model is valid upto 18 Gy per fraction.

[line 340] What does “the features LET” mean?

Comments on the Quality of English Language

[line 60] “In the latest years” could be “Recently.”

[line 264] “revael” must be “reveal”

[line 316] Check the grammar. “since …” is inappropriate here.

Author Response

Reviewer 3 Report (New Reviewer)

Comments and Suggestions for Authors

The authors studied using experimental ionizing radiation data of human cells treated, by previous published works, calculated normal and tumor cell survival α and β coefficients of the linear quadratic established model. Also they calculated complex DNA damages using the MCDS code, in order to complete any missing data in their model. The topic and aim is interesting, but there are some issues in the manuscript which must be revised.

General Comments:

1- What is the importance of this research compared to previous works? What does it add to previous research?

2- In this research, the previous results from 1980 to 2022 have been mentioned, and considering the submission of the article in 2024, it is better to review the results of 2023 and add them to the work.

3- MCDS code is used in calculating the damage of missing data. In my opinion, this code is a very good and fast, but according to the experimental results and the comparison of the results in this research with the experimental results (Figure 3) and recent articles with the Geant4-DNA code and the better proximity of these results with the experimental results, it is better the Geant4-DNA code was used.

4- How are the results of figure 3 calculated? normal cells, tumor cells, both normal and tumor cells, how is it simulated with the code? Also, the mentioned figure is not clear. Please draw it more clearly and include lines and symbols.

5- Because you work in microdosimetry sizes, it is better to mention the frequency-mean lineal energy and frequency-mean specific energy (y_f and z_f) quantities.

Round 2

Reviewer 3 Report (New Reviewer)

Comments and Suggestions for Authors

While thanking the authors for answering the referee's comments, please consider the following points in the text of the article according to the previous comments:

1- It is mentioned in the introduction of the manuscript (line 54) in the field of Monte Carlo methods. Considering the importance of DNA damage and its repair, it is suggested to refer to other codes such as:

PARTRAC like the following studies:

Friedland W and et al. 2017, "Comprehensive track-structure based evaluation of DNA damage by light ions from radiotherapy-relevant energies down to stopping," Sci. Rep. 7 45161

Geant4-DNA like the following studies:

Moeini H and Mokari M 2022, "DNA damage and microdosimetry for carbon ions: Track structure simulations as the key to quantitative modeling of radiation-induced damage," Med. Phys. 49 4823-4836

Mokari M, Moeini H and Soleimani M 2021 Calculation of microdosimetric spectra for protons using Geant4-DNA and a μ-randomness sampling algorithm for the nanometric structures Int. J. Radiat. Biol. 97 208–218

KURBUC like the following studies:

Nikjoo H, Taleei R, Liamsuwan T, Liljequist D and Emfietzoglou D 2016, "Perspectives in radiation biophysics: From radiation track structure simulation to mechanistic models of DNA damage and repair," Radiat. Phys. Chem. 128 3-10

2- Line 473 of the text of the article mentions the restorative model used in this research. It is suggested to check the results of the work and the used model according to the update of the repair model mentioned in the new works. As:

Mokari, M., Moeini, H., & Farazmand, S. (2023). Computational modeling and a Geant4-DNA study of the rejoining of direct and indirect DNA damage induced by low energy electrons and carbon ions. International Journal of Radiation Biology, 99(9), 1391-1404.

Author Response

We thank the reviewer for his/her positive assessment of our work and revisions. We are optimistic on the outcome of our last effort for addressing comments and suggestions by the reviewer.

1- It is mentioned in the introduction of the manuscript (line 54) in the field of Monte Carlo methods. Considering the importance of DNA damage and its repair, it is suggested to refer to other codes such as:

PARTRAC like the following studies:

Friedland W and et al. 2017, "Comprehensive track-structure based evaluation of DNA damage by light ions from radiotherapy-relevant energies down to stopping," Sci. Rep. 7 45161

Geant4-DNA like the following studies:

Moeini H and Mokari M 2022, "DNA damage and microdosimetry for carbon ions: Track structure simulations as the key to quantitative modeling of radiation-induced damage," Med. Phys. 49 4823-4836

Mokari M, Moeini H and Soleimani M 2021 Calculation of microdosimetric spectra for protons using Geant4-DNA and a μ-randomness sampling algorithm for the nanometric structures Int. J. Radiat. Biol. 97 208–218

KURBUC like the following studies:

Nikjoo H, Taleei R, Liamsuwan T, Liljequist D and Emfietzoglou D 2016, "Perspectives in radiation biophysics: From radiation track structure simulation to mechanistic models of DNA damage and repair," Radiat. Phys. Chem. 128 3-10

RESPONSE: Indeed, reviewer is right, that adding these references makes our work more complete. We added all of them, as proposed.

2- Line 473 of the text of the article mentions the restorative model used in this research. It is suggested to check the results of the work and the used model according to the update of the repair model mentioned in the new works. As:

Mokari, M., Moeini, H., & Farazmand, S. (2023). Computational modeling and a Geant4-DNA study of the rejoining of direct and indirect DNA damage induced by low energy electrons and carbon ions. International Journal of Radiation Biology, 99(9), 1391-1404.

RESPONSE:  The reviewer correctly suggests this comparison. For this reason, we added the following part in p.6: ‘Moreover, by comparing our results to these of [18], we may observe an agreement among the repair percentages. Most importantly, comparing the results from this work, and as seen in Figure 8 of the mentioned article, for 100keV electrons, that means for the majority of our collected data, their repair percentage (remaining lesions) at 2h time point for example is around 40%, while ours vary around 45-50%. This is considered in very good agreement and give us more confidence on the validity of our approach.

We also attach the history of review-rounds, as requested.

This manuscript is a resubmission of an earlier submission. The following is a list of the peer review reports and author responses from that submission.

Round 1

Reviewer 1 Report

Comments and Suggestions for Authors

In this manuscript, the authors present details of a radiobiological database they have developed and made available online (http://radbiodb.physics.ntua.gr/radphysbio/). This represents a very large amount of effort, and could be a valuable resource for the community. However, there are quite a number of issues I observe with the database which hamper its usability, and which I would recommend be addressed before a public release. I'll outline points about the database itself first, and then comment on the manuscript.

Database comments:

Experiment ID: Good unique IDs are provided, but it would be useful if possible to include the figure/table where the data is taken from - from a quick survey, in some cases this isn't immediately clear from looking at the related paper.

Cell Line: It looks like the authors have preserved the name of the cell line as reported in the paper here. While reasonable, this makes analysis challenging, as there are many variants (e.g. AG01522, AGO1522, AGO-1522, AG0-1522...) used for the same line. This would make subsequent analyses more difficult. I'd recommend changing the cell line to use a standard list (E.g. from Cellosaurus) or add a 'Consensus cell line' name column which has the standardised name alongside the reported name.

Contamination: Related to the above, many cell lines are now known to be contaminated by other cultures. For example, the HSG line has over 100 entries, but is now known to actually be a HeLa derivative (https://www.cellosaurus.org/CVCL_2517). As this mislabelling affects the identified site of origin, this then affects all subsequent analyses depending on this, like the authors' figure 1. It is important that when confirming culture names that any contaminated lines are correctly labelled and tissue types are updated to reflect this.

MCDS data: Unfortunately, it's not really clear to me what this data adds. MCDS can be used as a predictive tool in some cases, but the authors themselves show that it actually performs quite poorly in this dataset, and it adds a lot of columns whose meaning is not very clear at all. Likewise, using a model to impute dataset variables risks forcing the data to fit more closely to preconceived expectations, which is not what's wanted in a database like this. My recommendation would be to remove the MCDS data entirely, or at least move it all to a group at the end of the database, to more clearly distinguish between modelled and observed data, and not use it in any fits.

LQ Fit parameters: Approximately 10% of points have clearly unreasonable values (alpha or beta in the 100s or 1000s). These would heavily disrupt any attempt to use the database for model fitting or development (and it's unclear how the authors themselves avoided this issue in the text?). I would recommend the fitting methods be checked to determine why this is happening, and everything re-fit. Also see comment below on fitting method more generally.

Also, as some LQ fit parameters are as reported and some have been re-fit by the authors, it would be good to include a column about which is which.

DSB/Gy: I'd recommend splitting out the modality (f/p) from the numbers into a separate column to make it easier to parse.

Manuscript comments:

Line 68 on: Here, the authors mention the PIDE, and then go on to discuss databases which only contain a few data types. However, the PIDE and most similar databases do focus on different particle types, and do contain most of the other parameters also included in this work, so this text should probably be revised to reflect this.

Line 77 on: This text is a bit unclear. If a model is populated with MCDS data, then it will likely only reproduce or embed the MCDS predictions, which significantly reduces the value of such a database. I'd recommend sticking to raw experimental data as the first instance.

Table 1: I'd in general recommend against direct comparisons between in vivo and in vitro LQ parameters - in vivo responses are heavily regulated by tissue type and condition, and comparing with cancerous data is a risk. In addition, the tabulated values are largely based on X-ray data, so it's probably not appropriate to make against a comparison against all radiation qualities, and instead to focus on x-ray data too.

Figure 1: In addition to comments above, just plotting a single average bar here obscures the large amount of data going into this plot. I'd recommend a more data-rich plot (E.g. with individual data points, or a box-and-whisker plot). This would also permit the inclusion of the outliers in the plot.

Figure 2 & Table 2: R-square is a somewhat limited metric for very heterogeneous data such as this, as it's sensitive to outliers. It may be valuable to also calculate the Spearman correlation, which is less sensitive to such outliers. Or, alternatively, you can consider only lower LET values. Also, in this and subsequent figures, it may be worth considering putting LET on a log scale, due to the density of points clustered at very low LET.

Figure 3: The figure quality is quite poor here, I would recommend putting the axis label under the X-axis and beside the y-axis as in other plots, and perhaps log-scaling the X-axis as above.

In addition, this model suggests there's little relation between the reported and modelled DSB yield values, again raising the issue of if the MCDS values are a useful complement to the experimental data.

Figure 4 & associated text: The correlation here in Figure 4 is apparently dominated by a small number of outliers, so again I would recommend looking at alternative correlation metrics to better reflect the uncertainty at low and moderate alpha values. And in addition, a more direct performance metric such as Mean Absolute Percentage Error would be a useful guide to predictive performance.

And as above, it would be worth considering log-plotting, or expanding out the majority of points with alpha and beta <1. This should also perhaps be accompanied by a more in-depth critique of the model performance at those lower values.

Line 206: The LQ is not necessarily that 'mechanistic' - many early definitions were empirical, and it arguably only captures a small portion of the known mechanisms of radiation response.

Line 226: The comment that the LQ provides almost equal predictions to other models isn't really accurate - the defining feature of the LEM and MKM is that they predict the LET dependence of radiation responses, which is inherently not included in the LQ model. While the resulting trend is approximately LQ, these other models capture features not incorporated in the basic LQ.

Line 230: As noted, the comparison between in vitro and in vivo LQ values is likely not directly meaningful, as is doing it without consideration of uncertainties and variability in data.

Line 239: While the difference in alpha between different cell lines is less than that of the most extreme LET conditions, there is known to be quite significant differences amongst cell lines, particularly for low LET x-rays. Neglecting this would be a significant limitation in a predictive model, so I think it would be better to adjust the phrasing around cell type here.

Line 297: It's not clear that beta is poorly predicted just because of uncertainties. If it was purely uncertainty driven, one would have expected a broad spread in data, whereas in this data the model seems to be dramatically over-estimating the range of beta, which suggests some sort of more systematic fitting error, which could be explored in more value.

Line 307: The RMSE is a reasonable estimate of error, and it would be good to have some more critical analysis of this here. In particular, the quoted RMSE in alpha, 0.52, is greater than the median value of alpha in the database (0.32). This would suggest overall model performance is quite poor even for alpha, as the error is large compared to the values. Some more discussion of this would be useful.

Line 362, supplementary data 1: In the LQ fitting analysis an unweighted fit to the survival values is used. I would typically recommend fitting to the log-transformed data (or equivalently, absolute fit weighted by survival). Because of how survival studies are designed, and the nature of the LQ, errors are not uniformly distributed as a function of survival, with typically smaller (absolute) uncertainties at lower survival levels. If an absolute fit is used, it tends to over-fit to high survival levels and over-estimate cell killing at low survival. This may also reduce some of the other uncertainties and errors in the database parameters.

Line 378: As noted above, it's unclear if supplementing with synthetic data is appropriate if you then wish to fit new models to experimental data.

Line 392, section 4.2: This section on the DNA repair fitting seems somewhat out of place, as it's unclear what was actually done here - was the data refit to repair kinetics? On which data, as the database doesn't seem to contain kinetic data? And were the parameters used to inform anything else, as they are not presented or discussed in the main text. It would be useful if the motivation and application of this model could be clarified.

Line 425: This final comment about 'Netlify' reads more like advertising copy than any scientific information. I'd recommend removing that.

Line 449 on: The final text here discussing model design seemed to get broken up into a rather disordered structure towards the end, with new lines for some points but not all, and a few rather disconnected sentences. I'd recommend rewriting this in more complete sentences.

In addition, if I've read this correctly, it seems like the authors have done both fitting and benchmarking on the entire dataset. This often leads to significant performance over-estimation, and I'd recommend applying a suitable cross-validation technique to give a more robust estimation of model performance (which will in turn impact on Figure 4 above and associated discussions).

Reviewer 2 Report

Comments and Suggestions for Authors

The primary objective of this paper is to establish a database using literature-derived double-strand break (DSB) data. The authors have employed Monte Carlo simulations to compute DSB when unavailable in existing literature. Additionally, a machine learning model was leveraged to predict the alpha and beta parameters of the survival curve. While the primary goal of establishing a database using literature-derived double-strand break (DSB) data is commendable, this work has significant structural and writing issues.

I believe that the creation of this database holds considerable interest and utility for the scientific community. However, the paper requires substantial refinement and restructuring. It notably lacks a comparison with the PIDE database, and why the authors did not choose to extend the pre-existing database by incorporating DSB-related information is unclear. A clear explanation or justification for this choice should be included in the paper.

The origin of DSB values, whether from Monte Carlo simulations or experimental data, is inadequately clarified. There is also ambiguity regarding the calculation of Linear Energy Transfer (LET) and its origin. This raises concerns about the robustness of DSB calculation based on LET, especially when LET is determined from Relative Biological Effectiveness (RBE) in certain cases. The LET-RBE relationship should be at least accompanied by associated uncertainties.

The presentation of the machine learning model requires an important revision, as the utility of this model remains unclear. The predicted beta parameter is unsatisfactory, and the alpha parameter exhibits errors at least comparable to existing mechanistic models like LEM IV. Given the unsatisfactory beta prediction, it may be impractical to pursue eXplainable Artificial Intelligence (XAI). The unusual prominence of irradiation conditions in the beta prediction raises concerns. Additionally, I recommend using 70% of the database for training and 30% for testing to enhance prediction robustness. Separating the alpha and beta predictions is advisable, especially given the extremely bad beta prediction.

There are in the paper some irrelevant details, such as the mention of using Excel or WebPlotDigitizer to extract values. The plethora of toolkits employed, including Matlab, Python, WebPlotDigitizer, Origin, R, and Excel, can be perplexing for readers, and it remains unclear why many of these distinct software applications were used when they could potentially be replaced by a unified tool.

The discussion primarily serves as an introduction and lacks substantive content in the first half. The plots presented in the paper lack meaningful significance. For instance, Figure 2 reiterates information well-documented in the literature, while an analysis of dispersion regarding the linear fit could enhance its value. Figure 4 is missing units of measurement. Figure 6 can also benefit from more details and improvement.

I recommend heavily restructuring the paper to include a more detailed database analysis and further information about database values. The emphasis on the machine learning algorithm should be reconsidered unless it yields satisfactory results. In my opinion, the database, coupled with the Monte Carlo simulations, provides a sufficient basis for publication if presented effectively.

Comments on the Quality of English Language

The language used is often not up to the standards of a scientific publication. I recommend using software for the rewriting of the paper.

Round 2

Reviewer 1 Report

Comments and Suggestions for Authors

Unfortunately, I don't think the authors have meaningfully engaged with many of my concerns about the data quality in the database, which are of critical importance if this is to be presented as a resource to the community. I don't disagree that some of these changes may be time-consuming to implement, but that's a consequence of attempting to develop such an extensive resource.

As a result, I can't reasonably recommend this manuscript for publication. I'll re-emphasise the key issues below, and strongly encourage the authors to take these into account should they wish to resubmit this manuscript either here or elsewhere.

Data origin: While not as critical as the other points, I would recommend the authors ensure tracability of both the data (by figure, table, etc.) or fitting parameters. Firstly, this is important to provide confidence in the data, and secondly it is invaluable if people are trying to resolve ambiguitities - for example, while attempting to resolve issues with fitting parameters described below, I can't reconcile this with the referenced paper because they clearly do not match. So it's unclear what a user is to do to try and work out if they think they've identified an issue with the data. So while time-consuming, I'd recommend the authors review their data to provide such labels.

Experimental conditions: Not providing consensus cell line names and especially not documenting cell line contamination/mis-identification in the database is at best not recommended practice, and at worst actively harmful to follow-up analyses. Simply noting that a large, but un-specified, portion of the data in the database is known to be wrong is not an appropriate way to manage a data resource, as this will almost certainly be over-looked by many people using this resource - not least of all the authors themselves in subsequent analyses in this manuscript!

The claimed concern about ethics is specious - I cannot imagine an author protesting the inclusion of a consensus cell name and consensus tissue of origin column, and indeed any author who did protest this is likely wrong to do so.

MCDS data: I fully agree with the need for more DSB *data*, but the MCDS model prediction is not data. There are probably dozens of different DSB prediction algorithms which would give different results here, with various degrees of validation. The MCDS produces 'reasonable' DSB values, but it doesn't capture the true experimental variability, and if someone wishes to develop a new model the presence of the MCDS points is at best distracting, if not actively harmful. As it's solely a function of other data in the database, it's not an independent predictor, and can negatively impact on model performance. As a result, I'd strongly recommend removing it from the database, and focusing on a collation of actual experimental data, which is what the field most particularly needs.

Fitting parameters: I was not concerned about the slightly negative beta parameters, this is an entirely expected side-effect of LQ fitting in real data. I am concerned about the ~10% of the database with enormous fitting values (alpha or beta of 100 or 1000) which clearly do not match any data in the linked paper (e.g. entry 2223 has alpha = 9797 Gy^-1, 1629 has alpha =5140 Gy^-1, neither of which are anything like the referenced data). This would have huge impacts on any attempt to fit to this data, and indeed raises a question about how it was avoided in the manuscript analysis, and the fitting processes more generally, which should all be reviewed.

See also the comment related to line 362 - here, I am not speaking of any plotting approaches, but if the authors which to use an un-weighted fit, it is better practice to fit ln(S) = -alpha*D - beta*D*D. The LQ model implicitly assumes that cell survival is distriuted exponentially rather than linearly, and in particular cannot go below 0, and so this gives a more accurate reflection of curvature and less liklihood of 'miss' at high survival levels.

There are a number of other points I'd still particularly disagree with, which I'll note below:

Table 1 & Figure 1 - My major concern here is a comparison between in vitro mixed cell lines and in vivo normal tissue. These factors are only distantly related, and direct comparisons are often highly misleading. The offhand comment about 'as done in thousands of other papers' later in the text is actively misleading - in vitro parameters are to be compared to in vitro parameters, and in vivo to in vivo - cross comparisons must be highly caveated at best, and are inappropriate at worst. This is not at all mentioned where the data is presented.

Figure 1: I strongly disagree that this figure is 'comprehensive', as it's the least informative summary of this data. Even including standard deviation ranges would be a useful addition, to highlight this range. S9 is, I think much better, but it's unclear why the authors split them into (arbitary?) paired groups and why a slightly more condensed version of this plot would not go in the main manuscript.

Figure 3: I genuinely cannot understand how the authors claim the MCDS and experimental data are in any way compatible here. They strongly disagree in both trend and in many cases magnitude, and it is, to my eye, clear that substituting or supplementing one of these for the other would dramatically impact model predictions.

Line 297: It is absolutely clear that there is a systematic over-fitting of beta, since the predicted values are in almost all cases much greater than the observed ones. The origin of this may be somewhat unclear, but this seems very unlikely to occur solely by uncertainties in data alone. Notably the limited refit in S11B greatly reduces this issue, suggesting some factor to do with outliers, but again it is hard to say.

Line 307: It's concerning to hear the RMSE described as "... just a parameter that shows that if it has a value of ~0 (almost never achieved in practice) would indicate a perfect fit to the data. In general, a lower RMSE is better than a higher one. No direct comparison with mean alpha value can be made therefore". This is a significant misunderstanding of the interpreation of this parameter. If your errors are assumed to be normally distributed, then the RMSE is the standard deviation of the residual between your model estimate and observed value - or, put another way, the magnitude of the unexplained variance by your model. If the RMSE is similar in magnitude to the typical values in your data (I used the median, rather than the mean, as the mean is outlier-sensitive), then this means the unexplained variance is greater than the magnitude of your predictions, and could be subject to +-100% or more errors. This is obviously not ideal for a predictive model.

Also, many of the new figures seem to be screenshots from a desktop with the 'activate windows' watermark still superimposed. While I don't care about software usage habits, this looks very unprofessional, as properly exported figures would also be of much better quality.
    Unfortunately, I don't think the authors have meaningfully engaged with many of my concerns about the data quality in the database, which are of critical importance if this is to be presented as a resource to the community. I don't disagree that some of these changes may be time-consuming to implement, but that's a consequence of attempting to develop such an extensive resource.

As a result, I can't reasonably recommend this manuscript for publication. I'll re-emphasise the key issues below, and strongly encourage the authors to take these into account should they wish to resubmit this manuscript either here or elsewhere.

Data origin: While not as critical as the other points, I would recommend the authors ensure traceability of both the data (by figure, table, etc.) or fitting parameters. Firstly, this is important to provide confidence in the data, and secondly it is invaluable if people are trying to resolve ambiguities - for example, while attempting to resolve issues with fitting parameters described below, I can't reconcile this with the referenced paper because they clearly do not match. So it's unclear what a user is to do to try and work out if they think they've identified an issue with the data. So while time-consuming, I'd recommend the authors review their data to provide such labels.

Experimental conditions: Not providing consensus cell line names and especially not documenting cell line contamination/mis-identification in the database is at best not recommended practice, and at worst actively harmful to follow-up analyses. Simply noting that a large, but un-specified, portion of the data in the database is known to be wrong is not an appropriate way to manage a data resource, as this will almost certainly be over-looked by many people using this resource - not least of all the authors themselves in subsequent analyses in this manuscript!

The claimed concern about ethics is specious - I cannot imagine an author protesting the inclusion of a consensus cell name and consensus tissue of origin column, and indeed any author who did protest this is likely wrong to do so.

MCDS data: I fully agree with the need for more DSB *data*, but the MCDS model prediction is not data. There are probably dozens of different DSB prediction algorithms which would give different results here, with various degrees of validation. The MCDS produces 'reasonable' DSB values, but it doesn't capture the true experimental variability, and if someone wishes to develop a new model the presence of the MCDS points is at best distracting, if not actively harmful. As it's solely a function of other data in the database, it's not an independent predictor, and can negatively impact on model performance. As a result, I'd strongly recommend removing it from the database, and focusing on a collation of actual experimental data, which is what the field most particularly needs.

Fitting parameters: I was not concerned about the slightly negative beta parameters, this is an entirely expected side-effect of LQ fitting in real data. I am concerned about the ~10% of the database with enormous fitting values (alpha or beta of 100 or 1000) which clearly do not match any data in the linked paper (e.g. entry 2223 has alpha = 9797 Gy^-1, 1629 has alpha =5140 Gy^-1, neither of which are anything like the referenced data). This would have huge impacts on any attempt to fit to this data, and indeed raises a question about how it was avoided in the manuscript analysis, and the fitting processes more generally, which should all be reviewed.

See also the comment related to line 362 - here, I am not speaking of any plotting approaches, but if the authors which to use an un-weighted fit, it is better practice to fit ln(S) = -alpha*D - beta*D*D. The LQ model implicitly assumes that cell survival is distributed exponentially rather than linearly, and in particular cannot go below 0, and so this gives a more accurate reflection of curvature and less likelihood of 'miss' at high survival levels.

There are a number of other points I'd still particularly disagree with, which I'll note below:

Table 1 & Figure 1 - My major concern here is a comparison between in vitro mixed cell lines and in vivo normal tissue. These factors are only distantly related, and direct comparisons are often highly misleading. The offhand comment about 'as done in thousands of other papers' later in the text is actively misleading - in vitro parameters are to be compared to in vitro parameters, and in vivo to in vivo - cross comparisons must be highly caveated at best, and are inappropriate at worst. This is not at all mentioned where the data is presented.

Figure 1: I strongly disagree that this figure is 'comprehensive', as it's the least informative summary of this data. Even including standard deviation ranges would be a useful addition, to highlight this range. S9 is, I think much better, but it's unclear why the authors split them into (arbitrary?) paired groups and why a slightly more condensed version of this plot would not go in the main manuscript.

Figure 3: I genuinely cannot understand how the authors claim the MCDS and experimental data are in any way compatible here. They strongly disagree in both trend and in many cases magnitude, and it is, to my eye, clear that substituting or supplementing one of these for the other would dramatically impact model predictions.

Line 297: It is absolutely clear that there is a systematic over-fitting of beta, since the predicted values are in almost all cases much greater than the observed ones. The origin of this may be somewhat unclear, but this seems very unlikely to occur solely by uncertainties in data alone. Notably the limited refit in S11B greatly reduces this issue, suggesting some factor to do with outliers, but again it is hard to say.

Line 307: It's concerning to hear the RMSE described as "... just a parameter that shows that if it has a value of ~0 (almost never achieved in practice) would indicate a perfect fit to the data. In general, a lower RMSE is better than a higher one. No direct comparison with mean alpha value can be made therefore". This is a significant misunderstanding of the possible interpretation of this parameter.
If your errors are assumed to be normally distributed (which I believe is implicitly done in their fitting method, unless they’ve modified that without noting it), then the RMSE is the standard deviation of the residual between your model estimate and observed value - or, put another way, the magnitude of the unexplained variance by your model. If the RMSE is similar in magnitude to the typical values in your data (I used the median, rather than the mean, as the mean is outlier-sensitive), then this means the unexplained variance is greater than the magnitude of your predictions, and could be subject to +-100% or more errors. This is obviously not ideal for a predictive model.

Also, many of the new figures seem to be screenshots from a desktop with the 'activate windows' watermark still superimposed. While I don't care about software usage habits, this looks very unprofessional, as properly exported figures would also be of much better quality.

Comments on the Quality of English Language

NA

Reviewer 2 Report

Comments and Suggestions for Authors

My final recommendation was to reject the paper in its present form. The one week of work to try to solve my comments is not satisfactory. The majority of my comments have been diminished by just adding a few written sentences in the article, while major revisions were needed to be published. I think that this approach of doing something quick and easy is not the best one to better the quality of this paper.

Comments on the Quality of English Language

None